Re-examining the relationship between invasive lionfish and native grouper in the Caribbean

Valdivia Abel 1 abel.valdivia@unc.edu
Bruno John F. 1
Cox Courtney E. 1
Hackerott Serena 1
Green Stephanie J. 2
1 Department of Biology, The University of North Carolina at Chapel Hill , Chapel Hill, NC , USA
2 Department of Zoology, Oregon State University , Corvallis, OR , USA
Somers Michael
Electronic publication date: 2014 Apr 15
Publication date: 2014
Volume: 2
Electronic Location ID: e348
Received 2014 Feb 3; Accepted 2014 Mar 25
Copyright: © 2014 Valdivia et al.
Copyright year: 2014
Copyright holder: Valdivia et al.
License: This is an open access article distributed under the terms of the Creative Commons Attribution License, which permits unrestricted use, distribution, reproduction and adaptation in any medium and for any purpose provided that it is properly attributed. For attribution, the original author(s), title, publication source (PeerJ) and either DOI or URL of the article must be cited.
License URL: https://creativecommons.org/licenses/by/4.0/

Keywords: Biotic resistance, Lionfish, Invasive species, Coral reef, Grouper, Caribbean, Predation

Funding: National Science Foundation OCE-0940019 National Geographic Society Committee for Research and Exploration grant 8514-08 This work was funded in part by the National Science Foundation, the National Geographic Society Committee for Research and Exploration, the Royster Society Carol and Edward Smithwick Dissertation Fellowship (to AV), the Rufford Small Grants Foundation (to CEC), the David H. Smith Conservation Research Fellowship (to SJG), and the University of North Carolina at Chapel Hill. The funders had no role in study design, data collection and analysis, decision to publish, or preparation of the manuscript.

==============================
Biotic resistance is the idea that native species negatively affect the invasion success of introduced species, but whether this can occur at large spatial scales is poorly understood. Here we re-evaluated the hypothesis that native large-bodied grouper and other predators are controlling the abundance of exotic lionfish (Pterois volitans/miles) on Caribbean coral reefs. We assessed the relationship between the biomass of lionfish and native predators at 71 reefs in three biogeographic regions while taking into consideration several cofactors that may affect fish abundance, including among others, proxies for fishing pressure and habitat structural complexity. Our results indicate that the abundance of lionfish, large-bodied grouper and other predators were not negatively related. Lionfish abundance was instead controlled by several physical site characteristics, and possibly by culling. Taken together, our results suggest that managers cannot rely on current native grouper populations to control the lionfish invasion.

Introduction

Biotic resistance describes the capacity of native or resident species in a community to constrain the success of invasive species (Elton, 1958). While there are several examples of native species controlling invasive populations, especially invasive plants (Reusch & Williams, 1999; Mazia et al., 2001; Magoulick & Lewis, 2002; Levine, Adler & Yelenik, 2004; Mitchell et al., 2006), less clear are the ecological mechanisms that allow heterogeneous communities to resist invasion (Lockwood, Cassey & Blackburn, 2005; Melbourne et al., 2007), and whether these processes are strong enough to compromise invasion success on a large scale (Byers & Noonburg, 2003; Davies et al., 2005). Especially elusive is whether native predators or competitors can constrain the expansion of exotic predator species at large spatial scales (but see, deRivera et al., 2005). Although biotic resistance substantially reduces the establishment of invaders, there is little evidence that species interactions such as predation completely prevent invasion (Levine, Adler & Yelenik, 2004; Bruno et al., 2005).

The invasion of Pacific lionfishes (Pterois volitans and Pterois miles) into the Caribbean basin (Schofield, 2009) over the past ten years provides an example of biotic interactions within a system that have been unable to reduce exotic invasion at a regional scale (Hackerott et al., 2013). Lionfish have spread to every shallow and deep habitat of the Western North Atlantic and the Caribbean (Whitfield et al., 2007; Betancur-R et al., 2011) including fore reef and patch reef environments (Green & Côté, 2009; Albins & Hixon, 2013), seagrass meadows (Claydon, Calosso & Traiger, 2012), mangrove root forests (Barbour et al., 2010), estuarine habitats (Jud et al., 2011), and even depths of ∼90 m (S Green, pers. obs., 2013). Lionfish dissemination in the region has added additional stress (Albins & Hixon, 2013; Lesser & Slattery, 2011; Côté, Green & Hixon, 2013) to an already disturbed coral reef ecosystem (Paddack et al., 2009; Schutte, Selig & Bruno, 2010). Their voracious appetite threatens small reef fish and juveniles of depleted fish populations including commercially important species such as groupers and snappers, and keystone grazers such as parrotfishes (Albins & Hixon, 2008; Green et al., 2012; Green et al., 2013). The failure of the system to constrain invasion success may be associated in part to the lack of native predatory capacity due to overfishing (Carlsson, Sarnelle & Strayer, 2009; Mumby, Harborne & Brumbaugh, 2011), or weak biotic resistance by the native predators and competitors (Levine, Adler & Yelenik, 2004).

The first study to investigate the potential for biotic control of lionfish by native predators found an inverse relationship between the biomass of native groupers and lionfish on reefs at the Exuma Cays Land and Sea Park (ECLSP) in the Bahamas (Mumby, Harborne & Brumbaugh, 2011). Specifically, Mumby, Harborne & Brumbaugh (2011) found that grouper biomass could explain ∼56% of the variability in lionfish biomass, and concluded that large-bodied groupers can constrain lionfish abundance if a series of cofactors at the site level are kept constant (i.e., reef complexity, larval supply, habitat characteristics). To examine whether this relationship holds true at a scale that reflects the heterogeneity of Caribbean reefs, Hackerott et al. (2013) gathered data on lionfish and grouper abundance from 71 sites across multiple regions in the Caribbean. When accounting for several site-specific covariates, Hackerott et al. (2013) did not find a relationship between the abundance of lionfish and native predators/competitors at a broad spatial scale in the Caribbean.

Aside from the suite of variables considered by Hackerott et al. (2013), several other covariates that are known to affect fish community structure, but vary across the region, could mask the effect that native predators have on lionfish abundance. Accounting for spatial scale and potential cofactors is essential when evaluating the importance of any single variable in a spatial comparative study (MacNeil et al., 2009). In particular, fishing mortality, larval dispersal, habitat quality, connectivity, reef structural complexity, depth, ecological interactions, and a myriad of other factors control the population dynamics of reef fish species (Sale, 2002). Here we re-evaluated the relationship between large-bodied grouper and other predators and lionfish abundance, accounting for a broader set of covariates than those included by Hackerott et al. (2013) that may mediate the interaction between predators and the invader (Mumby et al., 2013). We also evaluated the grouper bio-control hypothesis proposed by Mumby, Harborne & Brumbaugh (2011) and provide new insights into how such biotic resistance is unlikely at the scale of the Caribbean reef system. The issue still remains how to best manage and/or reduce numbers of lionfish where they are currently found, and the only effective solution to date is direct removal by fishermen and divers (Barbour et al., 2011; Frazer et al., 2012; Green et al., in press).

Materials and Methods

Sites and fish surveys

Survey methods are explained in detail in Hackerott et al. (2013). In summary, we surveyed 71 coral reefs (3–15 m deep) across three distinct reef habitats (spur-and-grove, slope, and patch reef) in three regions of the Caribbean: The Bahamas, Cuba, and the Mesoamerican Barrier Reef (Belize and Mexico) from 2009 to 2012 (Fig. S1, Table S1). All these habitats were once dominated by the coral complex Montastraea/Orbicella (Edmunds & Elahi, 2007). Reef sites were selected to cover a wide range of reef fish abundance. To survey fish abundance, we conducted underwater visual censuses at each site using belt transects (for spur-and-grove and slopes) or roving survey dives (for patch reef) (see details in Hackerott et al., 2013). Fish biomass was calculated through the allometric length-weight conversion formula (Froese & Pauly, 2013) and scaling parameters for lionfish were obtained elsewhere (Green, Akins & Côté, 2011). Grouper was defined as the combined biomass of relatively large-bodied species such as Nassau (Epinephelus striatus), tiger (Mycteroperca tigris), black (Mycteroperca bonaci), and yellowfin grouper (Epinephelus intersticialis) as defined also by Mumby, Harborne & Brumbaugh (2011). These species could potentially prey on lionfish (Maljković, van Leeuwen & Cove, 2008; Mumby, Harborne & Brumbaugh, 2011) and are relatively more abundant than other potential predators in the region (Hackerott et al., 2013). Other predators considered in this study included any species that could potentially prey on lionfish (see Table S2 in Hackerott et al., 2013). To directly compare our study with the generality of the results by Mumby, Harborne & Brumbaugh (2011), we overlaid their values of fish biomass on our main biomass plot and added boxplots that described the distribution of both data sets.

Covariates

The site-specific parameters included as covariates in our statistical model were wind exposure, habitat type, protection status, depth, and time since invasion which are described in detail in Hackerott et al. (2013). We added two new variables to the models that are hypothesized to strongly modulate lionfish abundance (Mumby et al., 2013): human population density/reef area (humans/reef) which is a proxy for fishing effects (Newton et al., 2007; Mora, 2008), and is predicted to be negatively correlated with lionfish density; and reef complexity, which is a proxy for habitat heterogeneity within sites, predicted to have a positive effect on lionfish density (Green et al., 2012; Green et al., 2013). Human population density was calculated as the number of humans within 50 km (maximum number of people living within 50 km radius of each site). We chose 50 km because it is a reasonable range of human influence on Caribbean reefs (Mora, 2008). Estimates of human population counts for the year 2010 were obtained from the Gridded Population of the World V.3 at 0.25 degree resolution (SEDAC, 2010). Reef area was calculated within 10 km radius of each site, well below the average home range for certain predator species (Farmer & Ault, 2011). Reef area was calculated from the Global Distribution of Coral Reefs (2010) database as available at the Ocean Data Viewer (http://data.unep-wcmc.org/datasets/13). This database represents the global distribution of warm-water coral reefs compiled mostly from the Millennium Coral Reef Mapping Project (UNEP-WCMC et al., 2010). All spatial calculations were done in ArcGIS v10.0. Humans/Reef Area (humans/km2 of reef) was defined as:

Number of humans within 50 km/Reef area within 10 km/(π102) (km2)

To estimate reef complexity we used a rugosity index (0–5) estimated at the transect level, where “0” was a flat substrate with no vertical relief and “5” was an exceptionally complex substrate with numerous caves and overhangs (Polunin & Roberts, 1993). Relief complexity for Eleuthera and New Providence sub-regions was estimated by averaging measurements of reef height (i.e., the vertical distance between the lowest and highest point of the reef structure in cm), taken at five haphazard points within the survey area (either transect or rover diver area) (Wilson, Graham & Polunin, 2007). To make reef complexity estimates homogenous for all sites, we transformed the relief complexity estimates taken in Eleuthera and New Providence to the rugosity index, described by Polunin & Roberts (1993), by assigning a gradient of 0 cm to “0” and over 300 cm to “5”. This resulted in a continuous rugosity index for these two sub-regions that was comparable with the rest of the sites.

Data analysis

Before applying the statistical model, we explored the data and determined that a negative binomial or Poisson were the most plausible distributions for lionfish counts (Appendix S1). Additionally, we checked for collinearity among covariates. We ran a logistic regression model with all the covariates and examined the variance inflation factor (VIF) for each variable. We used a VIF > 2 as a threshold to determine collinearity (Graham, 2003). Depth was correlated with reef habitat type as shallower sites tended to be dominated by patch reefs. Thus we modeled these two factors separately. However, we found that keeping depth in the full model, together with habitat type, did not compromise fitting or the magnitude of the effects (Appendix S1).

We ran a generalized linear mixed-effect model using the Automatic Differentiation Model Builder (glmmADMB) package (Skaug et al., 2013) in R 3.0.2 (R Core Team, 2013). As the lionfish data were over-dispersed and with excess of zeroes (Hackerott et al., 2013), a glmmADMB which accommodates zero inflation was the most adequate model structure (Bolker et al., 2012). We modeled lionfish counts with a negative binomial type 1 distribution and log link because this model performed better than a Poisson distribution based on the Akaike Information Criterion (AIC) (Appendix S1). Since a negative binomial is a discrete distribution we included an offset in the model to account for survey area (sampling unit level), thus we could effectively analyze the relationship between the density of lionfish and grouper biomass, i.e.,: Log (LF Density)=Log (LF Counts)−Log (Survey Area)

Because lionfish density and biomass were highly correlated (Pearson’s product moment correlation ∼0.96, p < 0.0001, Appendix S1), the results of the model should be applicable to biomass as well. The rest of the covariates were considered fixed. We standardized and centered the numerical covariates to aid in comparison of the coefficient estimates. To account for spatial autocorrelation we nested sites within sub-regions and used them as random effects (see Table S1 for sub-regions). To validate the model we corroborated that no patterns were found on the plot of the model residuals versus fitted values.

Moran’s I similarity spline correlograms constructed from the residuals of the glmmADMB model (Zuur et al., 2009) graphically indicated that our mixed-effect modeling framework successfully accommodated the spatial autocorrelation observed in the raw data (Fig. S2). Additionally, we used Mantel tests (Mantel, 1967) to confirm the lack of spatial autocorrelation between the Pearson residuals of the model and the lag distance (in km) between sites (i.e., whether sites that are closer together were more similar), and found that the overall correlation coefficient for the model was low (r = 0.073, p = 0.0001). We performed the autocorrelation analyses using the spatial nonparametric covariance function (ncf) package version 1.1-5 (BjØrnstad, 2013). All analyses were performed in R version 3.0.2 (R Core Team, 2013). Additionally, we provide the entire workflow R code (Appendix S1) and the master data summary by site level (FigShare, http://dx.doi.org/10.6084/m9.figshare.899210).

Results and Discussion

Even when including proxies for fishing and habitat structure in our statistical model, we found no support for an effect of large-bodied grouper or other predator biomass on lionfish abundance (Fig. 1, Table S3). As in Hackerott et al. (2013), the effects of other covariates in our analysis (namely wind exposure, habitat type, and protection status) (Fig. 1) remained the principal factors that appear to influence lionfish abundance. Our analyses suggest that variation in lionfish density across the region is driven by environmental processes and human activity and not by biotic resistance from native predators.

Figure 1 Coefficient estimates (±95% confident intervals) showing the effect of different variables on lionfish abundance.

Lionfish counts were modeled with a generalized linear mixed effect model using the automatic differentiation model builder (glmmADMB) based on a negative binomial distribution type 1 and log link. Abundance values were obtained by adding the log of survey area as offset in the model. Numerical variables (top axis, circles) and categorical variables (bottom axis, squares) are on different scale for easy visual representation as the magnitude effects of the former are relatively smaller. For full summary of the model see Table S3.

The absence of a relationship between lionfish and native grouper biomass across a large scale suggests that the results of Mumby, Harborne & Brumbaugh (2011), which found a negative association across 12 sites—5 inside and 7 adjacent to a no-take reserve (ECLSP)—represented a subset of a much broader and complicated relationship driven by other factors (Figs. 1 and 2). The average biomass of large-bodied grouper in our study of the Caribbean region (7.6 ± 0.8 g m−2, mean ± standard error) was slightly lower (Wilcoxon test, W = 1197, p = 0.002) than that found by Mumby, Harborne & Brumbaugh (2011) at Exuma (10.0 ± 2.6 gm−2) (Fig. 2). In contrast, the average biomass of lionfish in our study (7.8 ± 0.5 gm−2) was ∼20 times higher (or ∼2 times higher excluding patch reefs, i.e., 0.7 ± 0.1 gm−2) than those found at Exuma (0.4 ± 0.1 gm−2) by Mumby, Harborne & Brumbaugh (2011) (Fig. 2). In that study, relatively low lionfish biomass (∼0.3 gm−2) was associated with relatively high grouper biomass (∼25 gm−2). However, across 71 sites in our study, lionfish biomass ranged widely (0–50 gm−2) at sites with equivalent grouper abundance (Fig. 2). Thus, while predators may negatively impact lionfish under a particular set of local conditions (Mumby, Harborne & Brumbaugh, 2011), the underlying relationship between lionfish and predator biomass was undetectable on a wide range of heterogeneous sites across the Caribbean region.

Figure 2 Relationship between mean grouper and lionfish biomass.

In this study, 71 fore reefs (black dots protected sites, grey dots non-protected sites) were surveyed and analyzed across the Caribbean. For comparison, we included 12 sites (red squares) surveyed at Exuma Cays Land and Sea Park by Mumby, Harborne & Brumbaugh (2011). The red fitted line is for the linear regression model by Mumby, Harborne & Brumbaugh (2011) that explain 56% of the variability of lionfish biomass due to grouper abundance. Note that red squares represent ∼16% of all sites. Boxplots are median (vertical or horizontal line), 50 and 90 percentiles for lionfish biomass (right) and grouper biomass (top). Boxplots with black dots (general mean) correspond to our study and boxplots with red squares (general mean) to Mumby, Harborne & Brumbaugh (2011). The empty circles are outliers. Axes are in log scale.

In this study, we assume that high predator biomass is indicative of high predatory capacity resulting from a high frequency of large individuals (Fig. 3A). Grouper at protected sites were, on average, larger (48.6 ± 1.5 cm TL, mean ± standard error total length) than those at unprotected sites (34.7 ± 1.1 cm) (t = −7.68, p < 0.001, Fig. 3A). It is unlikely that sites with relatively high grouper biomass have low predatory capacity as a result of more abundant, but smaller, individual fishes. Indeed, the exact opposite pattern is well documented in a wide range of habitat types for several fish species (Gust, Choat & McCormick, 2001; Friedlander & DeMartini, 2002; McClanahan et al., 2007). This seems to also be the case for groupers in our study (Fig. 3B). At sites with grouper biomass of at least 10 gm−2, which was the minimum biomass per site in the ECLSP (Mumby, Harborne & Brumbaugh, 2011), there were relatively high frequencies of medium/large individuals (Fig. 3B). Medium/large groupers (>30 cm TL) have been classified as having potentially high predatory capacity (Mumby, Harborne & Brumbaugh, 2011). We found relatively lower frequencies (<50%) of small individuals (<30 cm TL) across all protected sites. Therefore, it is unlikely that a lack of predatory capacity at sites with the highest grouper biomass (Figs. 2 and 3B) explains the absence of a relationship between lionfish and grouper in our results.

Figure 3 Histograms of grouper class size (total length in cm) by categories.

(A) Class size distribution for protected and non-protected sites, (B) for sites with over and under 10 gm−2 of grouper biomass, and (C) for reef habitat types. Note that over 90% of protected sites and sites with >10 gm−2 of grouper biomass have individuals >30 cm in total length. Only every other class size has a label for clarity.

While we did not find evidence for an effect of native predators on invasion status, lionfish biomass varied significantly between the reef types we examined. All of our fore-reef sites (slope and spur-and-groove) constituted high-profile habitats and we also included a set of patch reefs, a reef habitat common in the region. In particular, slope and spur-and-groove habitat had a negative effect on lionfish abundance (Fig. 1, Table S3) with higher average lionfish abundance in patch reef habitats (27.5 ± 2.1 gm−2 vs. 0.7 ± 0.1 gm−2). However, both lionfish and large-bodied grouper and predators were frequently observed in each of these habitats (Fig. 3C). The class size distribution for groupers among reef habitats were similar (Fig. 3C). Almost 90% of the patch reef sites had groupers in the 21–40 cm class size range, while ∼60% of slope and spur-and-groove sites had groupers within 31–50 cm total length (Fig. 3C). Although, the size distribution of our study sites indicates that grouper >30 cm TL (deemed ‘large-bodied’ by Mumby, Harborne & Brumbaugh, 2011) were frequently (over 50%) observed in patch reef habitats (Fig. 3C), we caution that other patch reefs across the Caribbean must be surveyed in order to make meaningful extrapolations of the observed patterns in this habitat.

Other variables may also partly explain the variability of lionfish abundance in the region. Wind exposure, specifically whether sites were located on the windward side, had a weak negative effect on lionfish abundance (Fig. 1). However, the mechanism behind this association is not well understood and a premature explanation may be misleading. Larval supply, which we did not measure, may contribute to the lack of biotic resistance. As with other reef fish species (James et al., 2002; Cowen & Sponaugle, 2009), differential larval supply could influence site-specific lionfish recruitment (Ahrenholz & Morris, 2010). However, such data are not available for our sites. While measuring larval supply would have been interesting, it was outside the scope of our study due to the large number of sites included and the regional scale of the analysis. Additionally, though larval supply can be predicted by biophysical models that describe oceanographic features such as wind direction, surface temperature, or tidal amplitude, these relationships are often taxon-dependent (Wilson & Meekan, 2001; Vallès, Hunte & Kramer, 2009).

The question from a management point of view is whether native predators can actually constrain lionfish abundance across the Caribbean, given the heterogeneity of the systems and the factors that seemingly affect lionfish abundance. While we found no evidence that large-bodied grouper or any other large-bodied predators influence lionfish invasion success across the region, this finding is expected based on other systems and examples of invasive predators. For example, there is weak support in the literature for the biotic resistance hypothesis of native species constraining exotic predators in natural ecosystems, and rarely can resident predators constrain the distribution expansion of the invader (Harding, 2003; deRivera et al., 2005). In fact, the exact opposite is typical in systems where native predators are abundant. For example, the successful invasion of the Burmese python (Python molurus bivittatus) in the Everglades of South Florida has not been constrained by potential and abundant predators such as alligators (Alligator mississippiensis) (Willson, Dorcas & Snow, 2011). Moreover, it is common that invasive predators feed on the juveniles of the resident predators and competitors (Snyder & Evans, 2006; MacDonald et al., 2007; Doody et al., 2009; Kestrup, Dick & Ricciardi, 2011; Willson, Dorcas & Snow, 2011; Côté, Green & Hixon, 2013), further weakening the potential resistance capacity of the system. Ecological interactions, such as predation and competition, seldom enable communities to resist invasion, but instead constrain the abundance of invasive species once they have successfully established (Levine, Adler & Yelenik, 2004). However, the abundance of lionfish across the region does not appear to be constrained by ecological interactions (Hackerott et al., 2013). In the one published record of grouper eating lionfish (Maljković, van Leeuwen & Cove, 2008), it could not be determined whether the lionfish were dead or alive when consumed. It is common for divers and tour operators to feed speared lionfish to native predators, including sharks (Busiello, 2011). However, there is no evidence that this practice has changed the natural predatory instincts of resident predators towards the invader and feeding speared lionfish to native predators is now being discouraged due to safety concerns for divers (Whittaker, 2013).

Our results indicate that protection status (i.e., whether sites were located within a marine reserve or not) also had a negative effect on lionfish abundance (Fig. 1). This is most likely due to targeted culling in protected areas. Morris & Whitfield (2009) suggested that lionfish removals should be focused on ecologically important areas, including marine protected areas and reserves. Lionfish removals have since occurred in many marine reserves through organized citizen programs (Biggs & Olden, 2011; López-Gómez, Aguilar-Perera & Perera-Chan, 2013) and by reef managers (J Cal, pers. comm., 2013). This effort is paying off and has the potential to greatly reduce lionfish abundance, at least temporarily (Barbour et al., 2011; Frazer et al., 2012; Côté, Green & Hixon, 2013). In our dataset, of the six sites with grouper biomass over 20 gm−2, five were in protected areas where culling is very likely occurring (Fig. 2). This pattern supports the results of our statistical analysis that lionfish abundance is reduced in marine protected areas due to some factor other than predator abundance. The negative effect of protection status on lionfish abundance and lack of effect of grouper or other predator biomass on lionfish abundance indicate that culling within protected areas most likely explains the observed pattern.

This analysis expands our original statistical model of the relationship between invasive lionfish and native grouper species (Hackerott et al., 2013) to include two additional covariates hypothesized to moderate the relationship between these species Mumby et al. (2013). After accounting for these additional processes, we find that: (a) the biomasses of lionfish and large-bodied grouper (or other predators) are not negatively related, and (b) lionfish biomass is controlled by a number of physical site characteristics, as well as by culling within marine reserves. Our study was motivated by the desire to explore whether the findings and solutions from local case studies will be effective elsewhere, which is key to informed management decisions about the invasion. We conclude that removals are most likely the only feasible mechanism for controlling lionfish at a Caribbean-wide scale.

Supplemental Information

Table S1 Detailed information of reef sites

Location names, coordinates, and site characteristics of surveyed sites. S&G, spur-and-groove.

Click here for additional data file.

Table S2 Summary of the glmmADMB results

Lionfish abundance (ind. 100 m−2) on grouper biomass (g 100 m−2), predators, and other co-factors.

Click here for additional data file.

Figure S1 Location of survey sites

For site abbreviations, surveys dates and coordinates refer to Table S1.

Click here for additional data file.

Figure S2 Moran’s I similarity spline correlograms for lionfish and grouper raw data across all sites (top two panels) and for the glmmADMB model residuals (bottom panel)

Note the strong spatial autocorrelation of the raw data (i.e., swirling lines around zero) and how the hierarchical structure of the random effects (sites nested in regions) of the full glmmADMB model eliminated this correlation in the model residuals. A Mantel test of the model Pearson residuals (r = 0.073) corroborates the lack of spatial correlation of the residuals. Lines are the mean ±95% confidence interval.

Click here for additional data file.

Appendix S1 R Code and main analysis

Click here for additional data file.

Supplemental Information 6 Lionfish versus grouper composite

Click here for additional data file.

We thank F Pina for logistic support in Cuba. C Layman, D Knowles, The Bahamas National Trust and Friends of the Environment provided logistic support in the Bahamas. We thank J Pawlik, M Marti, and L Deagan for their help and facilitation of a research expedition to Mexico. We thank the Belize Fisheries Department, the Southern Environmental Association, Healthy Reefs Initiative and the Toledo Institute for Development and Environment for support in Belize. We thank the Academic Editor of PeerJ, Michael Somers, and reviewers Charles Griffiths and Joseph Pawlik for commenting on this manuscript.

Additional Information and Declarations

Competing Interests

Author Contributions

Field Study Permissions

Data Deposition

The authors declare no competing interests. John Bruno is an Academic Editor for PeerJ.

Abel Valdivia conceived and designed the experiments, performed the experiments, analyzed the data, contributed reagents/materials/analysis tools, wrote the paper, prepared figures and/or tables, reviewed drafts of the paper.

John F. Bruno conceived and designed the experiments, performed the experiments, contributed reagents/materials/analysis tools, reviewed drafts of the paper.

Courtney E. Cox, Serena Hackerott and Stephanie J. Green performed the experiments, contributed reagents/materials/analysis tools, reviewed drafts of the paper.

The following information was supplied relating to ethical approvals (i.e., approving body and any reference numbers):

Bahamas: Department of Marine Resources, Ministry of Agriculture and Marine Resources. Permit MAF/FIS/17.

Cuba: Centro de Control y Inspección Ambiental, Cuba via Fabian Pina.

Mexico: Dirección General de Ordenamiento Pesquero y Acuicola de la Comisión Nacional de Acuicultura y Pesca (CONAPESCA) de la Secretaría de Agricultura, Ganaderia, Desarrollo Rural, Pesca y Alimentación (SAGARPA). Permiso DAPA/2/06504/110612/1608.

Belize: Belize Fisheries Department. Permit # 000028-11.

The following information was supplied regarding the deposition of related data:

Figshare: http://dx.doi.org/10.6084/m9.figshare.899210

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
