# Peer review of "Re-examining the relationship between invasive lionfish and native grouper in the Caribbean"

_PeerJ, doi:10.7717/peerj.348_

## Round 0.1 · original submission · Minor Revisions

I found this MS interesting and well written, and I have no issues with it. The one referee has some very minor comments (see below).

·

Basic reporting

This is a well written paper and other than a few editorial comments below I was happy with the presentation and usefulness of the paper

Experimental design

Good, there are plenty of replicates and a good experimental design

Validity of the findings

Useful and applicable findings for management of this invasive alien

Additional comments

there were a few edits needed to this otherwise useful and well written paper
line 26 use feet not m
line 124 data were ( pleural)
line 198 insert space between 30 and m
line 199 Fig. not Figure
line208 Data are ( pleural)
line 291 do not put sp name into italics ( as in all other refs)
line 295 do not capitalise words in titles of journal articles
line 310 b no vol of pages given to ref
line 210, 234, 247, 357, 414, 439, 445 do not capitise words in titles of journal articles
line 445 give journal in full

·

Basic reporting

All aspects of this article conform to basic reporting requirements.

Experimental design

All aspects of this article conform to experimental design requirements.
The experimental question and methods are clearly defined and appropriately analyzed.

Validity of the findings

The conclusions of this study are well supported by the data and analyses used in the article.

Additional comments

The authors are weighing-in on a topic of current controversy and importance in invasive species (and Caribbean coral reef) ecology by providing an additional analyses of a dataset used to examine the relationship between invasive lionfish and native predators. It is this reviewer's opinion that the scientific process is at its best when these controversies play-out with rapid publication that is unfettered by attempts to suppress valid but contrary analyses. For this system in particular, time will tell. The authors have done an excellent job of describing opposing studies in this article and are careful to point out weaknesses in the data and conclusions. The writing is clear and to the point, and was a pleasure to read.

---

## Round 0.2 · accepted · Accept

Well done on a polished MS.